# Computing hematopoiesis plasticity in response to genetic mutations and environmental stimulations

Yuchen Wen[1,2,3,*], Hang He[1,2,3,*], Yunxi Ma[1,2,3], Dengyi Bao[1,2,3], Lorie Chen Cai[3], Huaquan Wang[4], Yanmei Li[5], Baobing Zhao[6], Zhigang Cai[1,2,3,4,5]

**Cell plasticity (CP), describing a dynamic cell state, plays a crucial role in maintaining homeostasis during organ morphogenesis, regeneration, and trauma-to-repair biological process. Single-cell-omics datasets provide an unprecedented resource to empower CP analysis. Hematopoiesis offers fertile opportunities to develop quantitative methods for understanding CP. In this study, we generated high-quality lineage-negative single-cell RNA-sequencing datasets under various conditions and introduced a working pipeline named scPlasticity to interrogate naïve and disturbed plasticity of hematopoietic stem and progenitor cells with mutational or environmental challenges. Using embedding methods UMAP or FA, a continuum of hematopoietic development is visually observed in wild type where the pipeline confirms a low *proportion of hybrid cells* ($P_{hc}$, with bias range: 0.4~0.6) on a transition trajectory. Upon *Tet2* mutation, a driver of leukemia, or treatment of DSS, an inducer of colitis, $P_{hc}$ is increased and plasticity of hematopoietic stem and progenitor cells was enhanced. We prioritized several transcription factors and signaling pathways, which are responsible for $P_{hc}$ alterations. In silico perturbation suggests knocking out EGR regulons or pathways of IL-1R1 and β-adrenoreceptor partially reverses $P_{hc}$ promoted by *Tet2* mutation and inflammation.**

## Introduction

Hematopoietic stem and progenitor cells (HSCs and HPCs, or abbreviated as HSPCs) are primitive, multipotent cells capable of differentiating into various blood cell types, including both myeloid-lineage and lymphoid-lineage (1). In previous decades of research, it was suggested that HSPCs and their fates could be simply defined by their immunophenotypes and constituted a hematopoietic hierarchy; hence, a tree-like model was formulated for understanding hematopoiesis (2). The tree-like model aligned the fates of HSPCs and their downstream cell types in a discrete way: HSCs are at the top, whereas mature blood cells are at the bottom (3). In contrast, thanks to applications of single-cell omics, especially the single-cell RNA-sequencing (scRNA-seq) technology for profiling hematopoiesis, a continuous model for hematopoiesis was recently suggested (4, 5, 6, 7). Intriguingly, such continuous model echoes the "landscape" metaphor proposed by reference 8, saying that cell differentiation is a non-stop process like a ball rolling down a hill. The downward slide represents the cell differentiation. The ball at the high level turns to be unstable (high plasticity), whereas that at the low level of the landscape is with low energy and turns to be stable (low plasticity) (8). Although this kind of metaphor conceptually helps, a quantitative way rather than a descriptive way for technically understanding developmental trajectory is still lacking.

Cell plasticity (CP), describing a flexible and dynamic cell state (or fate; i.e., a cell could commit two cell types in the future and we define that cell as a hybrid cell), plays a crucial role in maintaining homeostasis during organ morphogenesis, regeneration, and damage-to-repair biological process, and even cancer-related tumorigenesis. However, currently there are few quantifiable methods to directly measure CP. Through profiling many cells with a high-dimensional matrix (i.e., scRNA-seq or CyTOF platform–based) along with lineage tracing techniques, currently it is feasible to measure CP in a more quantitative way (9, 10, 11, 12). And indeed, hematopoiesis represents one of the most important systems to investigate factors dictating CP because insights could be tested easily using ex vivo models, animal models, or transplantation assays (13, 14, 15).

Fine-tuned CP is a key to the development and maintenance of homeostasis in multicellular organisms, as well as in response to environmental disturbances (16). Generally, this process involves the activation of certain signaling pathways (SPs, cellular receptors

[1]National Key Laboratory of Experimental Hematology, Tianjin, China   [2]Tianjin Key Laboratory of Inflammatory Biology, Department of Pharmacology, School of Basic Medical Science, Tianjin Medical University, Tianjin, China   [3]The Province and Ministry Co-sponsored Collaborative Innovation Center for Medical Epigenetics, School of Basic Medical Science, Tianjin Medical University, Tianjin, China   [4]Department of Hematology, Tianjin Medical University Tianjin General Hospital, Tianjin, China   [5]Department of Rheumatology and Immunology, Tianjin Medical University Tianjin General Hospital, Tianjin, China   [6]Department of Pharmacology, School of Pharmaceutical Sciences, Cheeloo College of Medicine, Shandong University, Jinan, China

Correspondence: us36zcai@tmu.edu.cn
*Yuchen Wen and Hang He are co-first authors

**A**

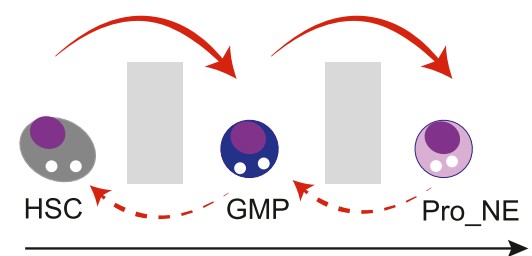

HSC　GMP　Pro_NE

Development trajectory

**Assumption #1**: Hybrid-cells have comparable probability of cell type A and B (*Prob.A* vs *Prob.B*) and accumulate at a transition stage (gray area);

**Assumption #2**: Proportion of hybrid-cells ( $P_{hc}$ ) represents the level of cell plasticity (CP) at the transition stage;

**B**

A discrete model

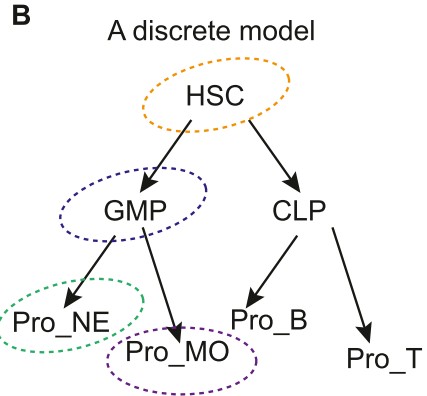

data from flow cytometry

A continuous model

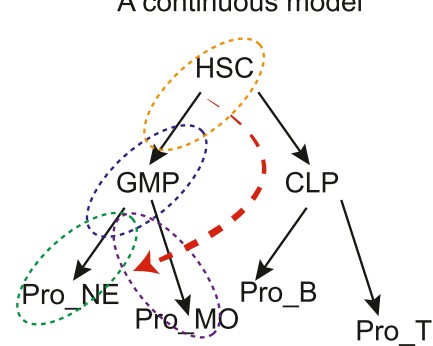

data from scRNA-seq

**C**

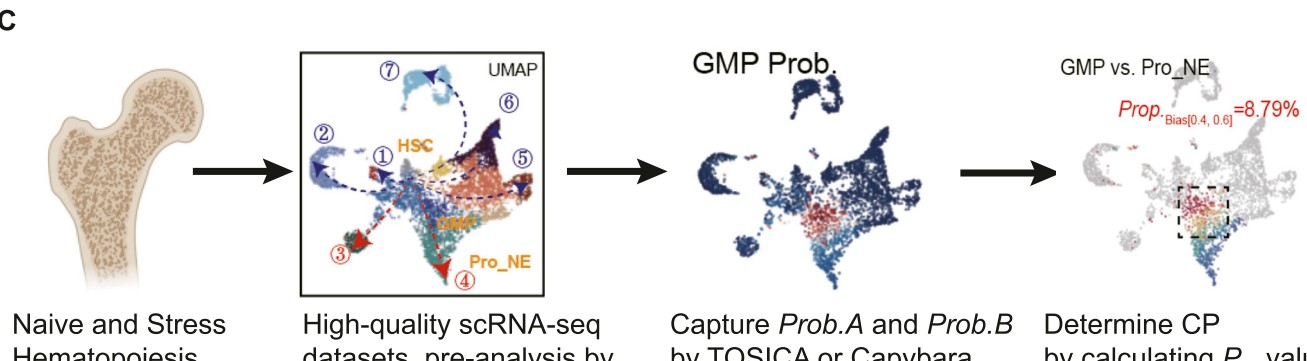

Naive and Stress Hematopoiesis

High-quality scRNA-seq datasets, pre-analysis by Seurat

Capture *Prob.A* and *Prob.B* by TOSICA or Capybara

Determine CP by calculating $P_{hc}$ value

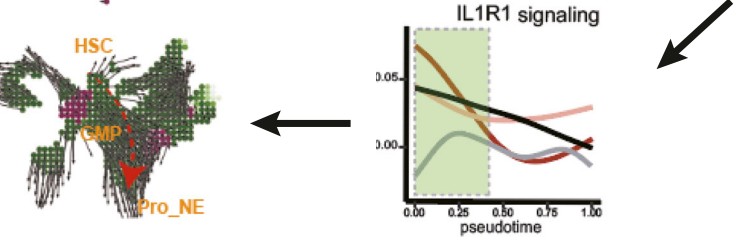

*in silico* perturbation of CP regulators

Prioritizing regulators of cell plasticity, including transcriptional factors (TFs) and signaling pathways (SPs)

that could be activated by environmental signals or circulating soluble cytokines) and transcription factors (TFs, regulating the organ morphogenesis [i.e., in solid organs] or compartment formation [i.e., in the hematopoietic and immune system]). For instance, numerous experimental works demonstrated that certain inflammatory signals, such as molecules like TNF-$\alpha$, IL-1$\beta$, and M-CSF, are effective in promoting HSC expansion (17, 18, 19, 20, 21, 22). Repeated direct activation of TLR4 by LPS (an inducer of inflammation) was shown to drive stationary HSCs into circulation and reduce their self-renewal capability (23, 24). Furthermore, it was reported that IFN-$\gamma$ treatment not only activated the proliferation of HSCs during infection (17, 25, 26), but also impaired the regeneration of HSCs by limiting their self-renewal potential (27, 28, 29). Alternatively, inhibition or overactivation of certain TFs will reprogram the cell fate–determining function and push the cells toward another destiny (30, 31, 32, 33, 34). For example, *HOXA3* and *HOXB6* are TFs related to the maintenance of HSC self-renewal, whereas *CEBPA* and *CEBPD* are TFs that direct HSC differentiation into neutrophil progenitors. In addition, as one of the important and common environmental factors, recent studies including ours suggest that inflammation affects the homeostasis and trajectory of HSCs (35, 36). Exact signaling pathways and transcription factors specifically dedicating HSPC plasticity remain largely unknown.

Kong et al developed a quadratic programming–based pipeline Capybara to capture hybrid cell states, but they did not quantify the portion of the hybrid cells along a certain trajectory (37). Here, we alternatively used a vision transformer (VIT)–based cell-type predicator TOSICA (38) and extracted cell-type probability values for quantitative analysis of cell plasticity. We introduced a parameter *proportion of hybrid cells* ($P_{hc}$; see Fig 1 and Results) on a transition trajectory to quantitatively detail CP. We suggest that a high $P_{hc}$ in a transition trajectory represents high cell plasticity for a cell or a pool of cells. $P_{hc}$ intuitively summarizes a property (or a state) for a cell (or a pool of cells) with susceptibility to deviate from its "current" identity and adopt an alternative destiny in the "future" (hybrid potential) (39). In brief, we established an analysis pipeline to effectively and quantitatively measure CP via computationally measuring $P_{hc}$ in naïve and stress hematopoiesis. We

named our vision transformer–based pipeline as scPlasticity (single-cell-omics–based plasticity analysis) and demonstrated it can visualize the continuous states and plasticity of hematopoiesis using mouse bone marrow lineage-negative (Lin⁻) cells.

## Results

### A quantitative way is required to recognize continuum and CP in hematopoiesis

In this study, we sometimes use a "cell type–like name" to represent a "cell state." As illustrated in Fig 1A, we made two assumptions to assist us to quantitatively analyze cell plasticity during cell-state transition:

Assumption #1: Hybrid cells have a comparable probability of cell types A and B (*Prob.A* versus *Prob.B*) and occur and accumulate at a transition stage/trajectory; for example, hybrid in-between cells probably accumulate between HSCs and GMP or between GMP and Pro_NE (Fig 1A).

Assumption #2: Proportion of hybrid cells ($P_{hc}$) represents the level of cell plasticity (CP) at the transition stage/trajectory; a high value of $P_{hc}$ means a high level of CP (Fig 1A).

Before the application of scRNA-seq analysis on hematopoiesis, naïve and stress hematopoiesis are generally analyzed by flow cytometry with limited cell surface markers (Fig 1B, left panel). Although the discrete tree–like model probably is right in principle, it cannot assist us to understand hybrid in-between cells during the hematopoietic development. In contrast, datasets from scRNA-seq or CyTOF technology provide a large number of observable parameters and assist us to recognize the continuum of hematopoiesis (Fig 1B, right panel), leading to generate novel insights for hematopoiesis and other developmental processes including understanding cell plasticity with transcriptomic inputs.

The overall working flow of the study, namely, the pipeline scPlasticity, is summarized in Fig 1C and with a brief introduction from Steps 1 to 5 in the Legend.

---

**Figure 1. Overall design of the study and a brief introduction of the ScPlasticity pipeline.**
**(A)** Cell states in a continuous development process could be simply divided into three stages: a start state (i.e., hematopoietic stem cells), an in-between state (i.e., granulocyte–macrophage progenitor), and an end state (i.e., neutrophil progenitors, Pro_NE here). **(B)** Hematopoiesis: a discrete model versus a continuous model. The discrete model was based on immunophenotype and transplantation analyses. Thanks to its simplicity, this classic model is still widely accepted (and probably right too). In contrast, the continuous model believes that during development, the commitment of a cell fate is stochastic and a continuum could be observed for the entire development course. The probability of a cell state (or a cell type) takes place at every stage of the development trajectory and could be mathematically captured using certain observable parameters. The continuous model appears to be more precise when a high-volume observable parameter is available and promote novel insights for hematopoiesis (dashed arrow in red). **(C)** Overall working flow of the present study. We generated scRNA-seq datasets for comparing the changes in cell compartments and plasticity under four scenarios. The scPlasticity pipeline is summarized as follows: Step 1: the input is a single-cell transcriptomic dataset (scRNA-seq). Of note, transcriptomic datasets are more straightforward than other modes of omics. The cell number should be greater than 1,000 to have a good performance; Step 2: the cells are annotated and visualized in a UMAP embedding or other embedding ways. Typically, we do UMAP embedding in Seurat and FA embedding in Palantir. Annotation of cells with Seurat recognizes cell states (or cell types) in a discrete way, and no probability parameters are provided; Step 3: the probability values for a cell as a cell state or a cell type are calculated using an alternative way (i.e., Capybara developed by Kong et al or TOSICA developed by Chen et al, both platforms were tested in the study; we only report the outcomes from TOSICA here as TOSICA with much better biological interpretation); Step 4.1: a biased value of a cell for cell fate A against cell fate B is calculated. The range of the bias value is between 0 and 1 when *Prob.A* is greater than *Prob.B*; Step 4.2: then, the value of proportion of hybrid cells ($P_{hc}$) is determined for a range of the biased value. We choose the range parameter 0.4–0.6 for calculating $P_{hc}$ as the cells within this range are naturally hybrid (having a very high probability to commit fate A or fate B); for the equations, see Fig 2E. Step 5: in the end, we use cumulative density plots (empirical cumulative density function), confusion matrix–like heatmaps (prediction versus reference), biased-value heatmaps, and values of $P_{hc}$ to recognize the plasticity of a cell (or a pool of similar cells) in a developmental process. See Fig 2 for the detailed outcome of Step 1 to Step 5. See https://github.com/cailab-bio/scPlasticity for our computational scripts.

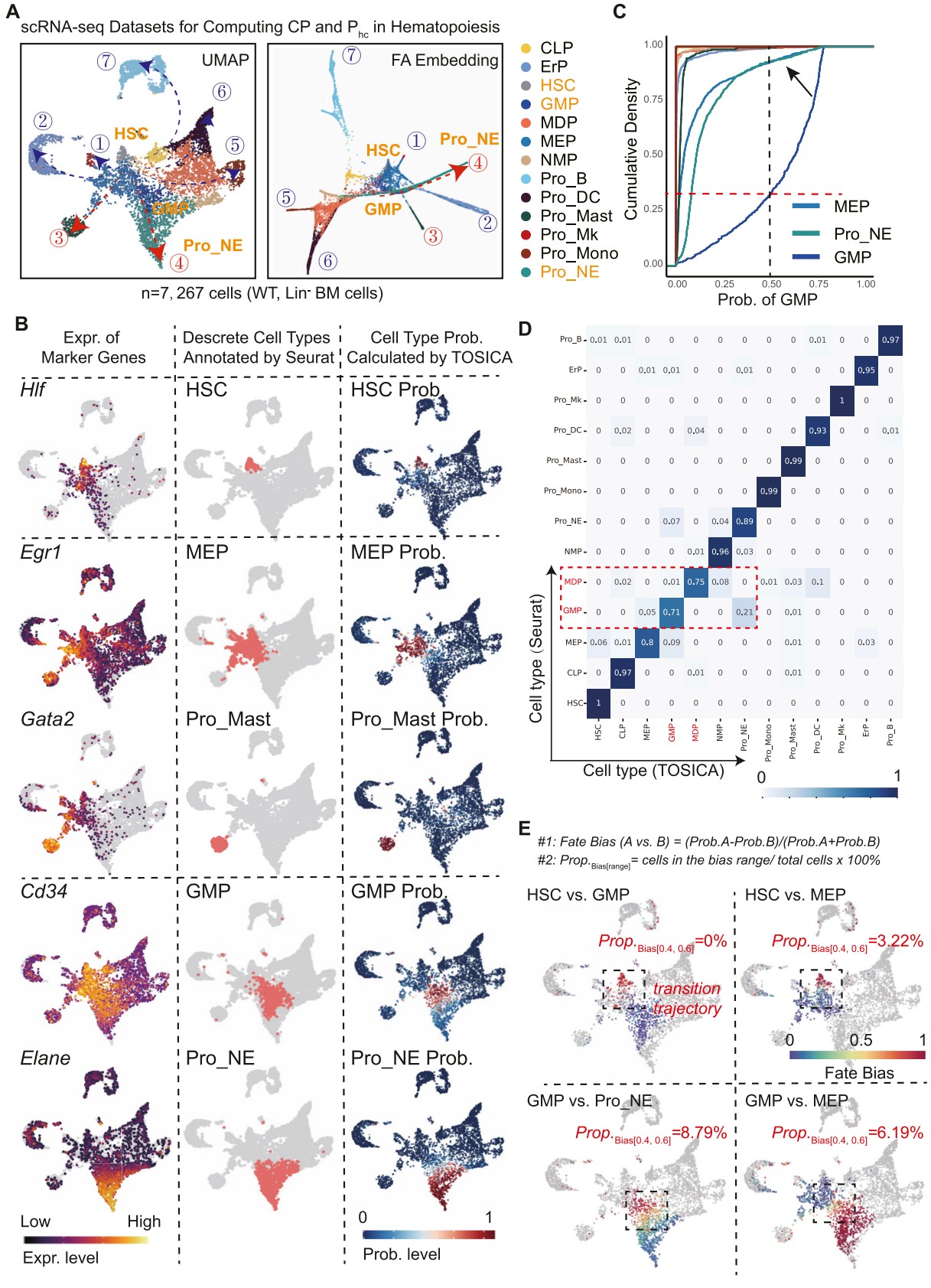

**A** scRNA-seq Datasets for Computing CP and $P_{hc}$ in Hematopoiesis

UMAP

FA Embedding

n=7, 267 cells (WT, Lin- BM cells)

CLP
ErP
HSC
GMP
MDP
MEP
NMP
Pro_B
Pro_DC
Pro_Mast
Pro_Mk
Pro_Mono
Pro_NE

**B**

Expr. of Marker Genes | Descrete Cell Types Annotated by Seurat | Cell Type Prob. Calculated by TOSICA

*Hlf* — HSC — HSC Prob.

*Egr1* — MEP — MEP Prob.

*Gata2* — Pro_Mast — Pro_Mast Prob.

*Cd34* — GMP — GMP Prob.

*Elane* — Pro_NE — Pro_NE Prob.

Low — High
Expr. level

Prob. level
0 — 1

**C**

**D**

Cell type (Seurat) vs Cell type (TOSICA)

**E**

#1: Fate Bias (A vs. B) = (Prob.A-Prob.B)/(Prob.A+Prob.B)
#2: $Prop._{Bias[range]}$= cells in the bias range/ total cells x 100%

HSC vs. GMP — $Prop._{Bias[0.4, 0.6]}$=0% — transition trajectory

HSC vs. MEP — $Prop._{Bias[0.4, 0.6]}$=3.22% — Fate Bias

GMP vs. Pro_NE — $Prop._{Bias[0.4, 0.6]}$=8.79%

GMP vs. MEP — $Prop._{Bias[0.4, 0.6]}$=6.19%

### Establishing a workflow for computing cell state and plasticity in naïve Lin⁻ cells

This study is a follow-up of our previous experimental research (40 *Preprint*), which studied the axis of bone marrow and gut by interrogating the role and mechanisms of clonal hematopoiesis in leukemia and colitis. We generated 4 scRNA-seq datasets of mouse bone marrow Lin⁻ cells across four distinct conditions (WT_Veh, Tet2$^{+/-}$_Veh, WT_DSS, and Tet2$^{+/-}$_DSS). After standard preprocessing, alignment, and quality control steps by Seurat, the remaining 7,267 bone marrow cells were obtained (Fig 2A). We visualized the dataset using the Uniform Manifold Approximation and Projection (UMAP) and Force Atlas (FA) approach and revealed 13 cell types. The continuum of cell embedding by UMAP and FA supports that our data are in a very good quality (Fig 2A). On the UMAP embedding, we delineated seven discernible endpoints of the hematopoietic development, including megakaryocytic progenitor (Pro_Mk), erythroid progenitor (ErP), mast cell progenitor (Pro_Mast), neutrophil progenitor (Pro_NE), monocyte progenitor (Pro_Mono), dendritic cell progenitor (Pro_DC), and B-cell progenitor (Pro_B) (Fig 2A).

Because the two developmental trajectories Pro_NE and Pro_Mast have demonstrated obvious changes in stress hematopoiesis (see Figs 3 and 4), we focus on these two branches for simplicity hereafter (Fig 2B, left and middle column). To capture the probability values of each cell (Seurat version 4 is unable to do that), we implemented a vision transformer–based cell annotation and predictor method TOSICA (38) to re-annotate cell types and extract the probability values (Fig 2B, right column). We found that there was a good match between the manually annotated cell types (by Seurat) and the cell types predicted by TOSICA (Fig 2B, middle and right column).

When using the cumulative density analysis (empirical cumulative distribution function [ECDF]–based plot; see the Materials and Methods section) for accessing the "ambiguity" of each cell type originally marked by Seurat, we found GMP cells appear to have the highest "ambiguity," an indicator of hybrid cells among the cell types (Fig 2C). When setting the threshold for the probability

value at 0.5 (vertical dashed line in black, Fig 2C), we found that ~30% of all GMP cells were with a probability value less than 0.5 (horizontal dashed line in red, Fig 2C). We used a confusion-matrix heatmap to systematically compare the overlapped portion of each cell type marked by Seurat and by TOSICA. Only the proportion values from megakaryocytic–erythroid progenitors (MEP), GMP, and monocyte–dendritic cell progenitor were less than 90% (Fig 2D), and only that of GMP and monocyte–dendritic cell progenitor is less than 80% (Fig 2D, dashed rectangle), suggesting most of the Lin⁻ cell types have relative low ambiguity at the unchallenged condition.

To further capture the degree of cell plasticity and proportion of hybrid cells ($P_{hc}$) on a transition trajectory, we used two equations to help us determine the fate bias and $P_{hc}$.

### *Equation #1*

$$Fate\ Bias\ (A\ Versus\ B) = (Prob.A - Prob.B)/(Prob.A + Prob.B).$$

Probability values of states A and B for a cell are from TOSICA metadata outputs. We use this equation to determine how faithful the cell marked as state A is committed to state A. When *Prob.A* is greater than *Prob.B*, the range of the cell fate bias is [0, 1]: 1 suggests that the cell has a definite state A, whereas 0.5 suggests that the cell has an equal probability and "hybrid" state between A and B. In addition, we use a heatmap to visualize the bias value of the cells subjected to the bias analysis on the same UMAP plot (Fig 2E; cells in gray are the backgrounds).

### *Equation #2*

$$N = No.\ of\ cells\ within\ the\ bias\ range.$$

$$M = No.\ of\ total\ cells\ for\ fate\ bias\ analysis.$$

$$Prop._{Bias\ [range]} = N/M.$$

---

**Figure 2.  ScPlasticity pipeline analysis on WT mouse Lin⁻ BM cells.**
**(A)** Two-dimensional (2D) embeddings by UMAP or by FA (Force Atlas) for WT Lin⁻ BM cells (HSPCs) according to the gene expression matrix of each cell. A total of 13 cell types are annotated and colored. In addition, a total of seven trajectory endpoints of the HSPCs are denoted (from #1 to #7): Pro_Mk, ErP, Pro_Mast, Pro_NE, Pro_Mono, Pro_DC, and Pro_B. **(B)** Expression of ground-truth marker genes for HSPC annotation (left panel; heatmap showing the gene expression level [log₂, low to high]), discrete cell types annotated by Seurat (middle panel; red dots for indicated cells and gray dots for background cells), and prediction of cell types by TOSICA (right panel; heatmap showing the probability values [Prob., 0–1]). For simplicity, only the clusters of HSCs, MEP, Pro_Mast, GMP, and Pro_NE and only two trajectory endpoints of the HSPCs are included in the study. The *WT* Lin⁻ BM cell dataset was used as the training dataset (reference). The parameters from the training step were reused for the following prediction studies. See the Materials and Methods section and tutorials of TOSICA (https://github.com/JackieHanLab/TOSICA) for details about implementing TOSICA and extracting metadata from the downstream computation in the study. See https://github.com/cailab-bio/scPlasticity for our computational scripts. **(C)** Cumulative distribution of each cell type for their possibilities of being predicted as GMP cells. A threshold (Prob. = 0.5) is shown by the vertical dashed line, marking a large portion (>0.75) of the predicted GMP cells manifest a probability greater than 0.5, whereas a large portion (>0.90) of the predicted MEP and Pro_NE cells manifest a probability of predicted GMP less than 0.5 (arrow). **(D)** Heatmap shows the proportion of cells in each row with cell types marked by Seurat (original label, shown on the left) is predicted as cell types marked by TOSICA (shown on the bottom). The value of each grid indicates the overlapped portion recognized by Seurat and by TOSICA. The y-axis indicates cells that are annotated by Seurat (of note, the sum of the possibilities of the same row is always equal to 1.00). The x-axis indicates cells that are annotated by TOSICA (sum of the possibilities of the same column is less or greater than 1.00, a plasticity indication of cell identity revealed by TOSICA). Of note, most of the 13 cell types have an overlapped grid value greater than 0.8, but such cross-values in MEP and GMP are less than 0.8, suggesting their fates should be recognized as "flexible" with relative high plasticity. **(E)** Calculation of the $P_{hc}$ value. We define hybrid cells are with a bias value 0.4–0.6; thus, the $P_{hc}$ is marked as *Prop._{bias[0.4, 0.6]}*. The cell fate bias is based on the probability metadata from the TOSICA-based query. The calculation equations are shown on the top of the panel. The heatmaps (gray dots are background cells) show the values of bias between HSCs versus GMP, HSCs versus MEP, GMP versus Pro_NE, and GMP versus MEP, respectively. The $P_{hc}$ value of the target cells with values of bias between 0. 4 and 0.6 is shown for each hybrid choice. Pro_Mk, Megakaryocyte progenitor cell; ErP, Erythrocyte progenitor cell; Pro_Mast, Progenitor Mast cell; Pro_NE, Progenitor Neutrophil; Pro_Mono, Progenitor Monocyte; Pro_DC, Progenitor Dendritic Cell; Pro_B, Progenitor B-cell.

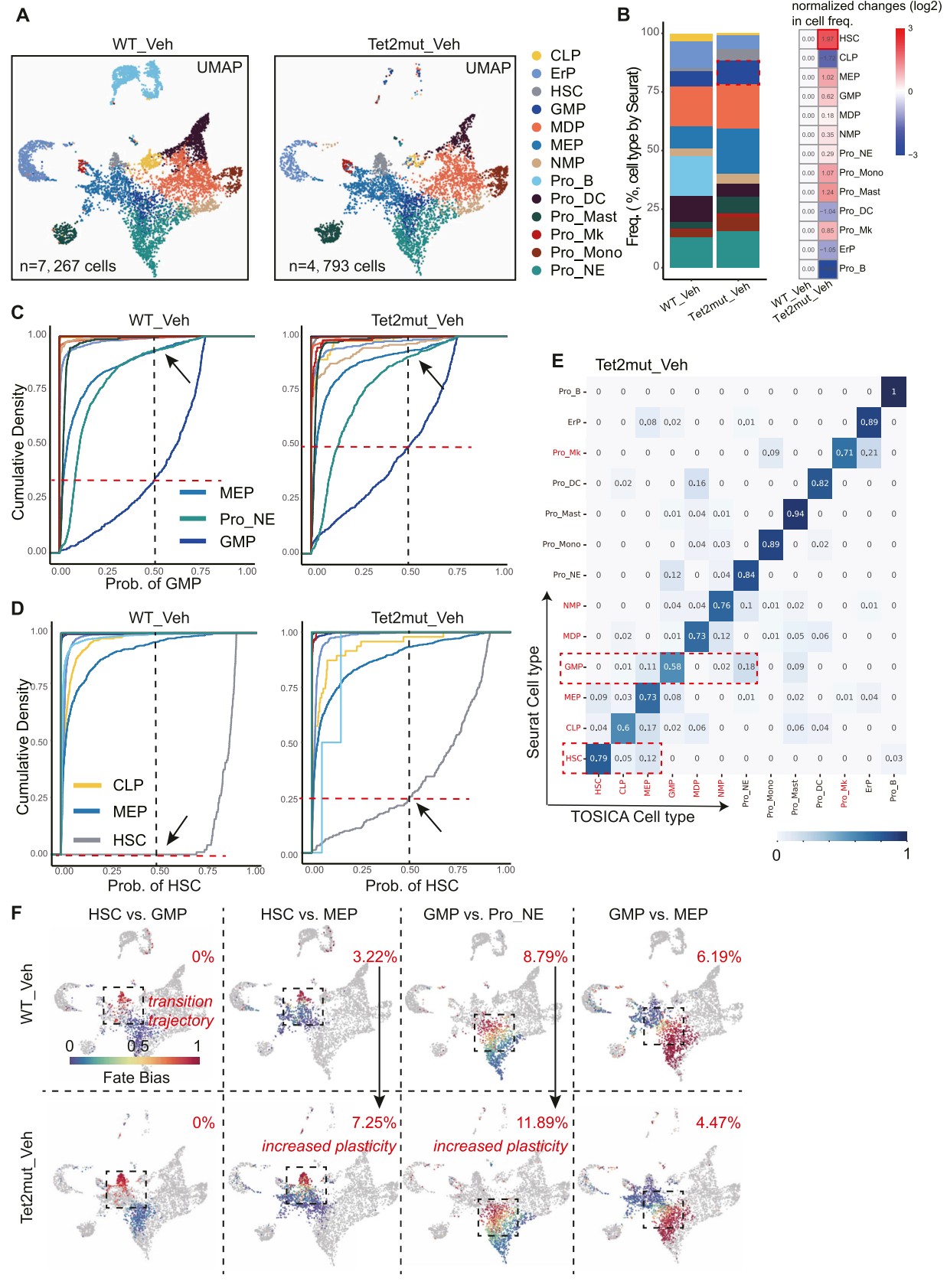

We define the cells within the range [0.4, 0.6] are faithfully "hybrid" cells. The proportion (Prop.) of these cells (thus here $P_{hc}$ = $Prop._{Bias\ [0.4,\ 0.6]}$) represents the plasticity level in the transition trajectory. We included the value of $Prop._{Bias\ [0.4,\ 0.6]}$ in the same UMAP plot (up-right corner). We also provide a table for the $Prop.Bias$ values with a range [0.35, 0.65] or [0.45, 0.55] (Table 1), suggesting the range [0.4, 0.6] is reasonable for our scenarios.

According to the equations shown above, we now can tell the developmental boundary between HSCs and GMP is very clear ($P_{hc}$ = $Prop._{Bias\ [0.4,\ 0.6]}$ = 0%), whereas that between GMP and Pro_NE is relatively ambiguous ($P_{hc}$ = $Prop._{Bias\ [0.4,\ 0.6]}$ = 8.79%) (Fig 2E).

In summary, by analyzing WT Lin⁻ BM cells, we used four statistical measurements: (1) ECDF plots, (2) confusion-matrix heatmap, (3) fate bias heatmap, and (4) the value of $P_{hc}$ ($Prop._{Bias\ [0.4,\ 0.6]}$) to quantify the hematopoiesis plasticity. Based on these measurements, we clearly tell GMP cells appear to be most ambiguous in cell fate choice and have the highest degree of cell plasticity under the naïve hematopoiesis condition, which is consistent with the empirical observations that GMP cells are a pool of very active cells in hematopoiesis.

### Tet2 mutation enhances plasticity of HSPCs

As reported by our previous studies among others, *Tet2* mutation is a well-known inducer of clonal hematopoiesis and may lead to chronic myelomonocytic leukemia–like diseases in aged mice. However, how *Tet2* mutation changed the landscapes of hematopoiesis remains largely unknown. We then used the same analysis pipeline scPlasticity shown above to investigate the impact of *Tet2* mutation (loss of function, LOF) on hematopoiesis plasticity. Employing the same stringent quality control criteria as previously outlined, we identified 4,793 cells in the *Tet2*mut_Veh group (Fig 3A). Results of cell proportions annotated by Seurat corroborated that *Tet2* deficiency enhances the preservation of HSC self-renewal potential (41, 42) (Fig 3B). Importantly, after the scPlasticity pipeline analysis on the *Tet2mut-Veh* scRNA-seq dataset, we demonstrated that the plasticity of HSCs and GMP is enhanced. As shown in Fig 3C, the proportion of the GMP cells with a probability less than 0.5 changes from 30% to 50%. Similarly, the proportion of the HSCs with a probability less than 0.5 changes from 0% to 25% (Fig 3D). The confusion-matrix heatmap suggests that 7 out of 13 cell types are with a shared proportion less than 0.8 in the *Tet2mut-Veh*, compared with that of 2 out of 13 cell types in the *WT-Veh* (Fig 3E, compared with Fig 2D). Accordingly, the value of $P_{hc}$ (HSCs versus MEP) is increased from 3.22% in the *WT-Veh* to 7.25% in the *Tet2mut-Veh*; that of $P_{hc}$ (GMP versus Pro_NE) increased from 8.79% in the *WT-Veh* to 11.89% in the *Tet2mut-Veh*. These measurements

demonstrate that *Tet2*-LOF induces changes in hematopoietic cell plasticity, with obvious effects observed in HSCs and GMP.

### Chronic colitis induced by DSS treatment leads to similar disturbed plasticity in hematopoiesis

To elucidate the changes in stress hematopoiesis challenged by the environment, we turned to perform similar scPlasticity pipeline–assisted CP analysis on *WT_DSS* and *Tet2mut_DSS* scRNA-seq datasets. As shown in Fig 4A and B, interestingly we observed the DSS also induces disturbed plasticity. The most dramatic abnormalities are observed in *Tet2*mut_DSS Lin⁻ cells. In the cumulative distribution plots, probabilities of GMP cells turn to be extremely ambiguous (the proportion of cells with probabilities less than 0.5 reach is as high as 0.75 in the *WT_DSS* group) (Fig 4C). Statistical analysis from the confusion-matrix heatmap and bias heatmap also suggests increased cell plasticity in the *WT_DSS* and *Tet2mut_DSS* group (Fig 4D–F). The values of $P_{hc}$ are also disturbed in the indicated cell fate choices (Fig 4F).

### Tet2 mutation enhances HSC self-renewal and exacerbates inflammatory response, whereas chronic colitis promotes myeloid differentiation

In addition to the statistical measurements, we are interested in master regulators of hematopoiesis plasticity. To elucidate the shared and distinct alterations in the three stress scenarios and discover underlying mechanisms behind *Tet2* mutation and DSS-induced colitis disorders, we compared the gene expression pattern and enriched pathways between the groups. For the comparison between the *Tet2*mut_Veh and the *WT_Veh*, we observed that the *Tet2*mut_Veh group exhibited significant up-regulation of inflammation-related genes (*Ifitm1*, *Ifitm3*, *Ifi27l2a*, *Ifi211*) and notable increases in HSC stemness-related genes (*Egr1*, *Nr4a1*) (Fig 5A). Furthermore, compared with *WT_DSS*, some chemokine-related genes (*Cxcl2*, *Ccl9*, *Ccl4*, *Cxcl2*) were also significantly enriched in *Tet2*mut_DSS (Fig 5B). Conversely, genes favoring mature cell populations exhibited significant down-regulation (Fig 5B). In addition, compared with the *WT_Veh* group, the *WT_DSS* group showed significant up-regulation of myeloid genes, particularly those related to neutrophil development, such as *S100a8*, *S100a9*, and *Ngp* (Fig 5C), consistent with previous studies suggesting serum levels of S100a8 and S100a9 are potential biomarkers for inflammatory bowel disease (43). Using the trajectory computational tool Palantir, we simulated the developmental trajectory from HSCs to Pro_NE. As shown in Fig 5D, the expressions of *Cd34*, *Il1r1*, *Egr1*, *Ly6a*, *Ifitm1*, and *Ifitm3* in *WT_DSS* and *Tet2mut_DSS* groups are significantly increased during the early to middle stages of

**Figure 3. Mutation in *Tet2* results in enhanced plasticity in the pool of HSCs and GMP cells.**
**(A)** UMAP embeddings of BM Lin⁻ cells from WT_Veh and *Tet2*mut_Veh mice. **(B)** Stacking bar plots showing the fraction of each cell type. Fold changes (normalized to the fractions in WT) are also presented (right panel, log₂). **(C)** Cumulative distribution of each cell type for their possibilities of being predicted as GMP cells. Three cell types are highlighted: MEP, Pro_NE, and GMP. **(D)** Cumulative distribution of each cell type for their possibilities of being predicted as HSCs. Three cell types are highlighted: HSC, MEP, and CLP. **(E)** Heatmap shows the proportion of cells in each row with cell types marked by Seurat (original label, shown on the left) is predicted as cell types marked by TOSICA (shown on the bottom) in the BM Lin⁻ dataset from *Tet2*⁺/⁻ mice. The value of each grid indicates the overlapped portion recognized by Seurat and by TOSICA. **(F)** Calculation of the indicated cell fate bias in the BM HSPC dataset from *Tet2*⁺/⁻ mice. The values of $P_{hc}$ are shown at the up-right corner of each panel. Note the $P_{hc}$ values of HSCs versus MEP and GMP versus Pro_NE are increased in *Tet2*⁺/⁻_Veh, an indication of increased cell plasticity (arrows).

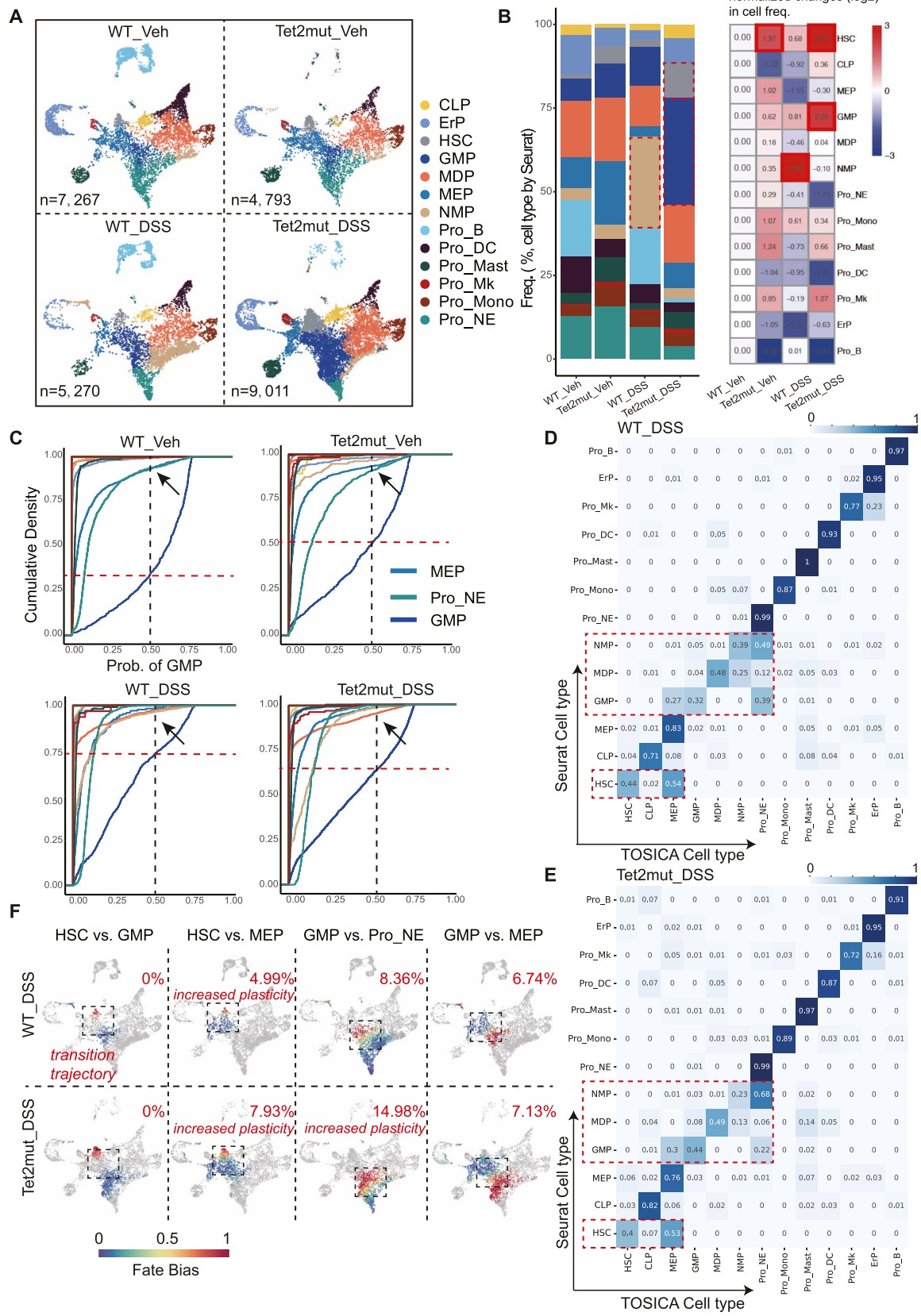

**Table 1.** $P_{hc}$ outcome test for the three bias ranges [0.35, 0.65], [0.4, 0.6], and [0.45, 0.55].

| Group | Bias ranges | HSCs versus GMP | HSCs versus MEP | GMP versus MEP | GMP versus Pro_NE |
|---|---|---|---|---|---|
| WT_Veh | [0.35, 0.65] | 0% | 8.27% | 9.91% | 12.97% |
| | [0.4, 0.6] | 0% | 3.22% | 6.19% | 8.79% |
| | [0.45, 0.55] | 0% | 2.26% | 3.93% | 3.66% |
| Tet2mut_Veh | [0.35, 0.65] | 0% | 10.25% | 8.26% | 17.67% |
| | [0.4, 0.6] | 0% | 7.25% | 4.47% | 11.89% |
| | [0.45, 0.55] | 0% | 3.91% | 2.64% | 6.19% |
| WT_DSS | [0.35, 0.65] | 0% | 7.71% | 10.33% | 13.45% |
| | [0.4, 0.6] | 0% | 4.99% | 6.74% | 8.36% |
| | [0.45, 0.55] | 0% | 2.95% | 3.29% | 3.27% |
| Tet2mut_DSS | [0.35, 0.65] | 0% | 12.18% | 10.82% | 22.60% |
| | [0.4, 0.6] | 0% | 7.93% | 7.13% | 14.98% |
| | [0.45, 0.55] | 0% | 3.94% | 3.72% | 8.07% |

hematopoiesis. Of note, the expression of *Il1b* in the *Tet2*mut_DSS group increased abruptly in the later stage of the Pro_NE developmental trajectory, an indicator of strongest alterations found in *Tet2mut*_DSS (Fig 5D).

When paying attention to the different consequences caused by gene mutation in *Tet2* and chronic inflammation by DSS treatment, we noticed that the hematopoietic abnormalities caused by *Tet2*-LOF mainly involve in abnormal activation of the IFN signaling pathway (44) and TGF-β signaling pathway (Fig 5E and F), whereas that caused by DSS-induced chronic colitis is mainly associated with the activation of the TNF-α (45) (Fig 5G and H). Our analysis also suggested that IL-1R1, IFN-α, TNF-α (TNFR), and β-AR (adrenergic receptor beta, ADRB) pathways were up-regulated at early stages of hematopoiesis during the stress scenarios; among them, Lin⁻ cells from *Tet2mut*_DSS appear to have the most dramatic changes (Fig 5I). Taken together, Lin⁻ cells from *Tet2mut*_Veh and *WT*_DSS manifest not only some shared alterations, especially in enhanced inflammation, but also distinct patterns (HSC self-renewal activity versus downstream differentiation).

### HSC self-renewal regulating transcription factors are potential regulators of stress-related plasticity

To identify abnormalities of transcription factors in the stress hematopoiesis induced by *Tet2* mutation and DSS, we performed computational analysis using pySCENIC (46).

After rounds of filtering, comparison, and prioritization, we noticed that several regulators of HSC self-renewal are concurrently up-regulated in both *Tet2mut*_Veh and *WT*_DSS Lin- cells. As shown in Fig 6A–D, both Egr1 expression and its regulon activity are increased in HSC, GMP, and MEP pools. We performed pseudotime analysis and observed the positive regulators of HSC maintenance, including Erg(+), Egr1(+), and Meis(+), are indeed aberrantly activated in all of the three conditions of stress hematopoiesis (highest levels appear in the condition of *Tet2mut*_DSS, Fig 6E), consistent with previous experimental findings (42, 47, 48, 49). We also validated that more than half of the downstream targets of Egr1 and their gene signature score (UCell score) displayed aberrant activation along the pseudotime axis of the disturbed hematopoiesis (Fig 6F–H).

### In silico depletion of the HSC self-renewal–related transcription factors manifests a potential to enhance HSC differentiation

We took advantages of the two well-validated computational tools, CellOracle and TOSICA (with a slight modification; see the Materials and Methods section), to perform in silico perturbation analysis (Fig 7A). The algorithm CellOracle combines gene regulatory network inputs and scRNA-seq datasets to do in silico perturbation modeling with technically precise performance (50). As an independent third party, we have validated the performance of CellOracle in a myeloid malignancy MDS in one of our recent studies (51). As shown in Fig 7B, perturbing *Egr1* (Egr1 knockout) in all of three stress conditions leads to increased multilineage differentiation. Similarly, perturbing each of the four transcription factors (*Egr2*, *Erg*, *Hlf*, and *Meis1*, also in silico knockout here), which are required for HSC self-renewal and up-regulated in the Lin⁻ cells of *Tet2mut*_DSS, results in expedited differentiation of neutrophils and other myeloid cells (Fig 7C). These simulation results suggest that the transcription factors prioritized for disturbed hematopoiesis

**Figure 4. Chronic colitis induced by DSS treatment results in a dramatic alteration of cell plasticity in HSCs and GMP.**
**(A)** UMAP embeddings of BM Lin⁻ cells from WT and *Tet2⁺/⁻* mice (*Tet2mut*) with or without DSS treatment. Cell numbers are shown for each scRNA-seq dataset. **(B)** Stacking bar plots showing the fraction of each cell type. Fold changes (normalized to the fractions in WT) are also presented (right panel, log₂). **(C)** Cumulative distribution of each cell type for their possibilities of being predicted as GMP cells. Three cell types are highlighted: MEP, Pro_NE, and GMP. **(D, E)** Heatmap shows the proportion of cells in each row with cell types marked by Seurat (original label, shown on the left) is predicted as cell types marked by TOSICA (shown on the bottom) in the BM Lin⁻ dataset from WT_DSS and from *Tet2⁺/⁻*_DSS mice. The value of each grid indicates the overlapped portion recognized by Seurat and by TOSICA. **(F)** Calculation of the indicated cell fate bias and the value of $P_{hc}$ (at the up-right corner) in the HSPCs from WT_DSS and from *Tet2⁺/⁻*_DSS mice.

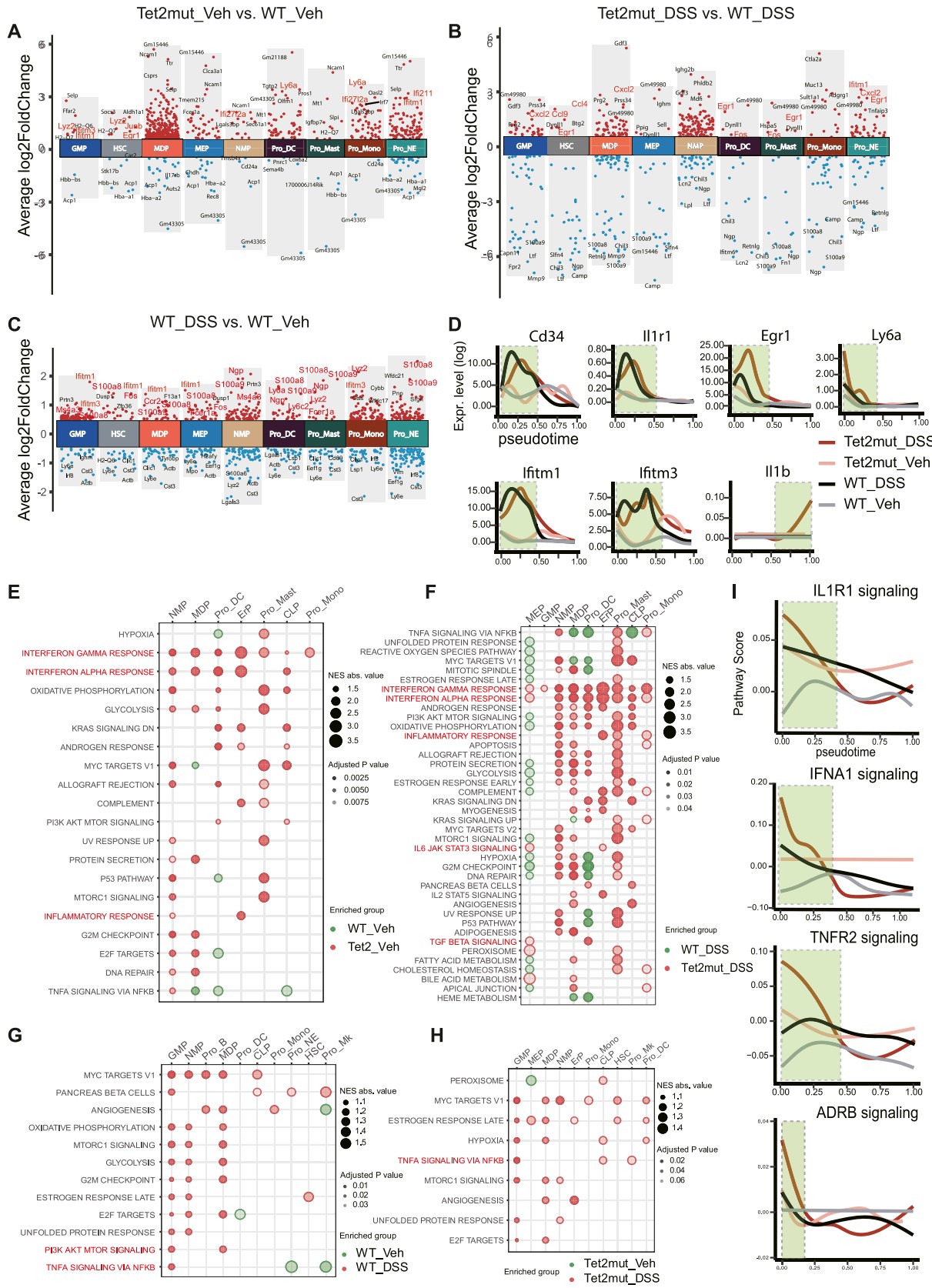

plasticity could be cross-validated by the CellOracle computational platform.

### In silico perturbation of interleukin-1 and adrenoceptor signaling pathways partially reverses the disturbed hematopoiesis

As shown in Fig 5I, we prioritized four signaling pathways that are dramatically up-regulated in the stress hematopoiesis: IL-1R1–, IFNA1-, TNFR2-, and ADRB-mediated pathways. Recently, several pieces of experimental studies including ours demonstrated that intervention of IL-1 signaling mitigates the *Tet2* mutation–induced aberrant hematopoiesis and associated chronic inflammation (i.e., chronic colitis) (40 Preprint, 52, 53). We aimed to perform CellOracle-like in silico perturbation to assess and empower the simulation capability of our Lin⁻ datasets. As the current version of CellOracle (version 0.18) unfortunately falls in shortness of using gene regulatory network other than transcription factors, we made a modified TOSICA (mTOSICA) analysis pipeline to achieve the prediction task: to predict changes in cell fate probability and cell-type proportions by simply permutating the expression of the genes in the certain pathway to a null value 0 (Fig 7A, lower panel). As shown in Fig 7D, the proportion of Pro_NE cells is subtly decreased upon the perturbation of IFN, ADRB, and IL-1R1 signaling pathways (from 15.8% to 15.0%, 15.0%, and 15.1%, respectively); however, the overall changes are quite minimal. We hypothesize that the changes in the proportion of hybrid cells ($P_{hc}$) would be more sensitive than the proportion of the cell type in the signaling-perturbation analysis. We captured the values of $P_{hc}$ and did a similar cumulative density analysis. Although the reversed effect is still partial, as shown in Fig 7E, 9 out of the 12 perturbation tests achieved the anticipated results. Similarly, using cumulative density statistical analysis, we also observed a partial reversal in the perturbation with knocking out of IL-1R1 and ADRB pathways (Fig 7F). Taken together, our mTOSICA pipeline is able to predict the anticipated consequences of knocking out signaling pathways with a fair accuracy.

## Discussion

As mentioned in Introduction, because the studies of hematopoiesis using cutting-edge lineage tracing technologies or scRNA-seq emerged a decade ago (54, 55), the accumulating datasets enable researchers to analyze the transcriptome and genealogies of thousands of cells at a single-cell resolution (56). Implementation of these technologies has generated large datasets and assisted us to unravel the cellular complexity of biological samples, especially in terms of subpopulation composition and their relationships (57).

With regard to computational tools, more than 100 algorithms have been developed to predict cell fate and trajectory, and methods could be grouped into three categories according to the distinct inputs: (1) the first one is the pseudotime-based cell trajectory approach, where a developmental trajectory is constructed based on dynamic changes in gene expression; the well-known methods include Monocle (58, 59) and Palantir (60); (2) the second approach is RNA velocity–instructed (61 Preprint, 62, 63, 64), which is recently developed and used to characterize cell dynamics by analyzing the ratio between the abundance of non-spliced precursor and spliced mature mRNA transcripts; however, interestingly approaches based on RNA velocity typically failed to accurately predict hematopoietic trajectory (i.e., computational analysis with the classic velocity or with scVelo (61 Preprint, 62, 63, 64)); and (3) the third approach is to use single-cell technology and combine with lineage tracking technologies (with implementing barcodes in stem cells or tracking mitochondrial mutation naturally); this approach may provide direct evidence about the trajectory of cells but technically challenging (56, 65, 66, 67).

In addition to the computational framework, this study also holds certain value for exploring the regulators for the hematopoietic development process. We applied scRNA-seq to analyze Lin⁻ cells (rather than LSK cells or HSCs) from mouse bone marrow, providing a valuable practice to choose appropriate cell inputs (according to our own experience for choosing Lin⁻ cells or LSK cells or HSCs). We investigated the plasticity changes induced by clonal hematopoiesis–related gene mutations and further explored cell-state alterations caused by environmental stimuli. Thus, both naïve and stress hematopoiesis were covered in the present study.

A cell at the transition trajectory may ambiguously commit two or more cell fates. We reason that such cells must exhibit mixed characteristics of two or more types. In addition to Capybara by Kong et al and scPlasticity by ours, several other studies attempted to understand this transition process (68, 69, 70, 71). Specifically, the Shannon entropy platform for cell–cell variance and the "Mellon" platform for cell-state density were reported recently to measure cell dynamics (72, 73). To our knowledge, few computational tools like our scPlasticity pipeline were reported to directly measure the degree of cell plasticity in hematopoiesis. In a recent report, Dussiau et al introduced the Shannon entropy to measure cell-to-cell variability among human hematopoietic cells during differentiation and demonstrated a dynamic stochastic process takes place at a transient stage of cellular indetermination, further suggesting hematopoietic cell plasticity acts at even normal

**Figure 5.  Alterations of the signaling pathways in stress hematopoiesis.**
**(A, B, C)** Manhattan plots showing the differentially expressed genes (DEGs) in each cell type in the comparison between *Tet2⁺/⁻*_Veh versus WT_Veh (A), between WT_DSS versus *Tet2⁺/⁻*_DSS (B), and between WT_DSS versus WT_Veh (C). Only DEGs with log₂FoldChange > 0.5 and *P* < 0.01 are shown. The top five DEGs are denoted. **(D)** Expression of critical genes along the developmental time (from HSCs to Pro_NE). The pseudotime and gene expression level were simulated by Palantir. Four scenarios are included in the same time axis: WT_Veh, Tet2⁺/⁻_Veh, WT_DSS, and *Tet2⁺/⁻*_DSS. The genes *Cd34*, *Il1b*, *Il1r1*, *Egr1*, *Ifitm1*, *Ifitm3*, and *Ly6a* are included. Except Il1b for marking maturation of monocytes, other genes have been suggested to be required for HSC maintenance and self-renewal activity. **(E, F, G, H)** Altered signaling pathways in the indicated comparison: (E) *Tet2⁺/⁻*_Veh versus WT_Veh; (F) *Tet2⁺/⁻*_DSS versus WT_DSS; (G) WT_DSS versus WT_Veh; and (H) *Tet2⁺/⁻*_DSS versus *Tet2⁺/⁻*_Veh. The circle size denotes the enrichment score, and color intensity denotes the adjusted *P*-value (*Padj*). NES, normalized enrichment scores. **(I)** Activity of pathways along the developmental time (from HSCs to Pro_NE). Inflammation- and stress-related pathways are included. IL-1R1, interleukin-1 receptor 1; IFNA1, interferon alpha; TNFR2, tumor necrosis factor receptor 2; ADRB, adrenergic receptor beta.

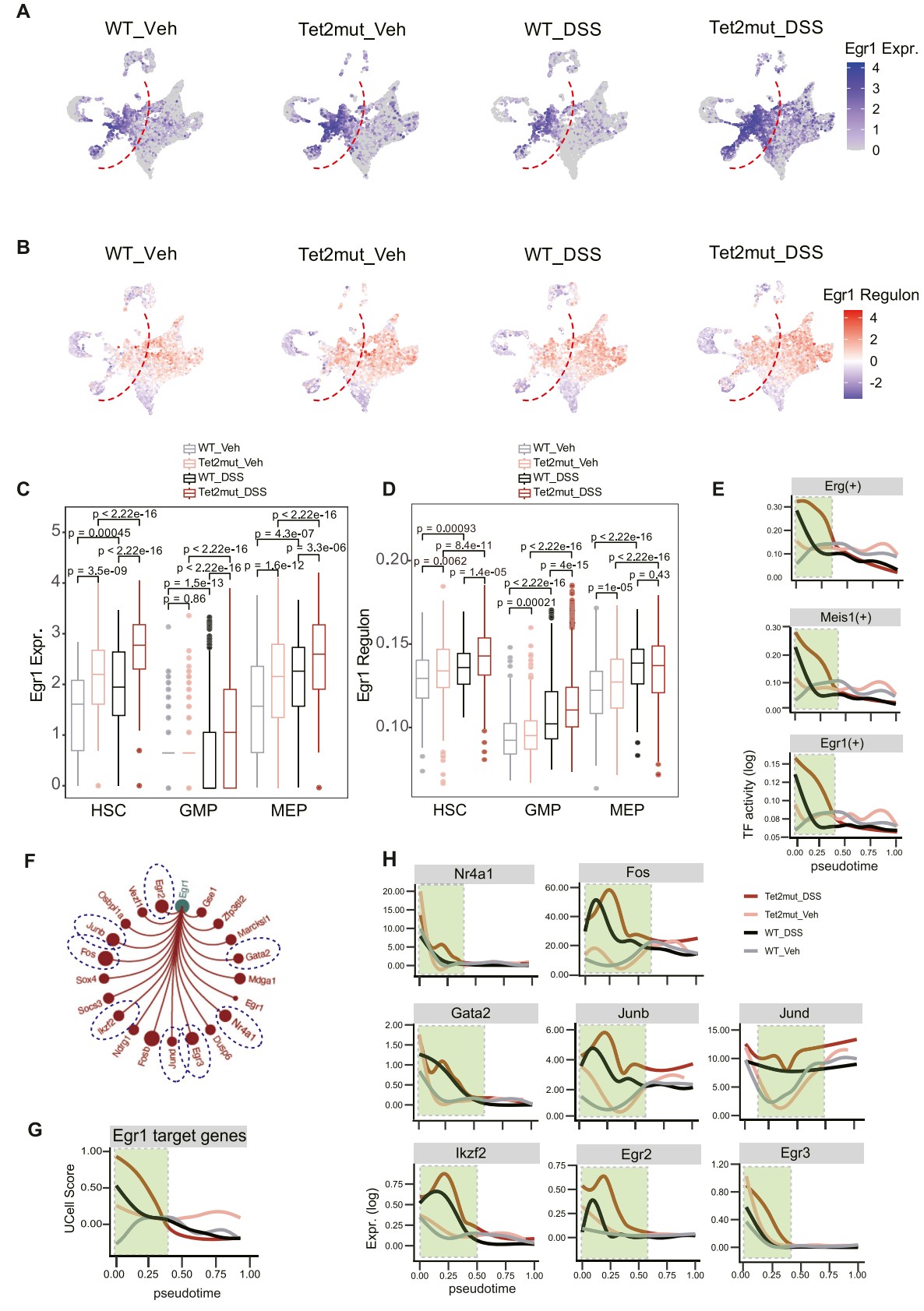

condition and disturbed cell plasticity at the diseased condition (72). It would be also worthwhile to use the "Mellon" platform to measure the changes in cell-state density introduced by *Tet2* mutation and DSS treatment.

Single-cell technologies offer unprecedented opportunities to explore heterogeneity at the individual cell level in biological processes. Although various methods have been applied to identify cell types, only a few quantitative approaches, such as SLICE (74) and SCENT (71) based on entropy measurements, have focused on single-cell differentiation. A new unsupervised computational framework, scRCMF (75), simultaneously identifies cell subpopulations and transition cells, quantifying transition cells from single-cell gene expression data using two measures: single-cell transition entropy (scEntropy) and transition probability (scTP). scEntropy measures cellular plasticity, whereas scTP predicts the fate decision and dynamic behavior of transition cells by calculating their probability of moving from the transition state to other states. Plasticity is crucial for restoring homeostasis after tissue damage, inflammation, or senescence but can also contribute to tumorigenesis (76). Yang et al used the scEffective-Plasticity score (12, 77), based on the Fitch–Hartigan algorithm (78, 79) and phylogeny tree–based distance, to mathematically assess the effective plasticity of each tumor cell. It would be worthy to compare our pipeline with these new tools in the future. In addition, it would be interesting to use the scPlasticity platform to measure the changes in cell plasticity in human diseased bone marrow cells such as MDS (51).

Pseudotime analysis, like using Monocle (58, 59), Slingshot (80), and Palantir (60), is important for scRNA-seq analysis. In the study, we applied Palantir to infer cell trajectories (60). We obtained pseudotime for all cells from the HSC to Pro_NE process, sorting cells according to their pseudotime order to show gene expression and activity changes. A recent advancement, GeneTrajectory (81), was introduced to directly infer gene trajectories from scRNA-seq datasets. It would be also worthy to implement this trajectory tool to dissect the hematopoiesis plasticity in the future.

In the study, we also prioritized several signaling pathways and transcription factors that likely dictate the plasticity dynamics. Simulating alterations of signaling pathways and transcription factors validated their involvement in regulating HSPC plasticity. Using scRNA-seq inputs, the study recognized the plasticity of naïve and disturbed hematopoiesis in a systematic and quantitative way and provided important molecular candidates for future experimental validation. We found that transcription factors *Egr1*, *Erg*, and *Meis1* were up-regulated after *Tet2*-LOF, and these transcription factors were also closely related to HSC self-renewal (42, 47, 48, 49).

In summary, we report a feasible and easy-to-follow quantitative analysis framework, the scPlasticity pipeline, for analyzing hematopoiesis plasticity under normal or stress conditions. In addition, we prioritized the transcriptional and signaling changes potentially responsible for the disturbed hematopoiesis plasticity. Finally, in silico perturbations of transcription factors and signaling pathways show that perturbations of the prioritized transcription factor *Egr1* lead to extensive enhancement of downstream differentiation, and that in silico perturbations of signaling pathways partially perturb cell states.

# Materials and Methods

## Generation of scRNA-seq datasets

scRNA-seq datasets were generated in our previous study and have been pre-analyzed by the typical Seurat pipeline (He et al, in press (82) or see the preprint version on BioRxiv (40 *Preprint*)). Four different scenarios were described in the previous and present studies: *WT*_Veh, *Tet2*⁺ᐟ⁻_Veh (or labeled as *Tet2mut*_Veh in Figures), *WT*_DSS, and *Tet2*⁺ᐟ⁻_Veh (or labeled as *Tet2mut*_DSS in Figures).

## Cell clustering, annotation, and visualization by Seurat

The data analysis was mainly performed in R (4.2.0) and Linux OS, and the in-house computing platforms have been described (83, 84). Standard preprocessing and quality controls were performed on the dataset using Seurat (version 4.4.0). Dimension reduction and visualization of the datasets were achieved through UMAP. To address batch effects, Harmony (version 1.2.0) was used for dataset integration. Clusters were identified using Seurat's cluster-finder computation algorithm, whereas cell types were annotated based on the expression of canonical tissue compartment markers.

## Cell-type annotation and collection of cell-type probability values by TOSICA

To infer the cell type with probability values, we used the vision transformer (VIT)–based transfer-learning tool TOSICA (38), which annotates cell types of the query dataset fast by transfer-learning reference dataset with a good benchmark in interpretability and accuracy. Here, we used *WT*_Veh BM Lin⁻ cells as a reference dataset to train a GOBP pathway–masked TOSICA model and applied the model to predict the cell type of each cell in WT and the other three conditions. Once the data frame with cell type (Seurat-annotated) and cell-type probability (TOSICA-annotated), we use the value in the dataframe to generate the heatmap matrix and use the "stat_ecdf ()" function from the ggplot2 to create ECDF plots.

**Figure 6. Transcription factor *Egr1* appears to be a shared hallmark mediating the disturbed plasticity in hematopoiesis.**
**(A)** Expression of *Egr1* on the UMAP plots. **(B)** Regulon activity of *Egr1* on the UMAP plots. **(C)** Quantifying expression levels of *Egr1* in HSCs, GMP, and MEP. **(D)** Quantifying regulon activity of *Egr1* in HSCs, GMP, and MEP. **(E)** Regulon activity of *Erg*, *Meis1*, and *Egr1* across the four groups along the pseudotime. The pseudotime from HSC to Pro_NE development is computed with Palantir. **(F)** Downstream genes regulated by *Egr1*. Most of the genes are related to HSC self-renewal. **(G)** Signature score of *Egr1* target genes (UCell score) along the pseudotime. **(H)** Expression of indicated Egr1 downstream targets along the pseudotime.

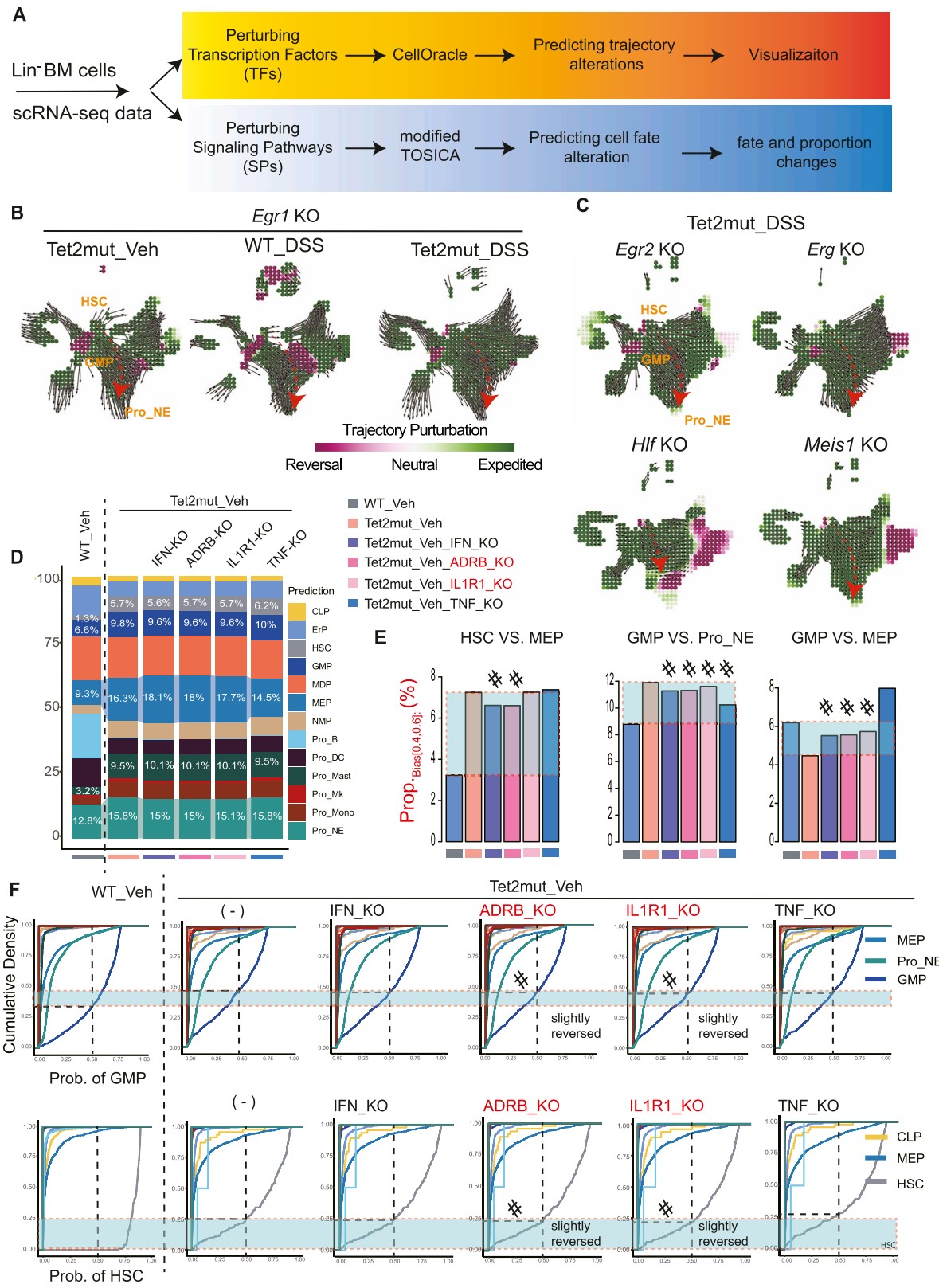

**Figure 7. In silico perturbations of transcription factors and signaling pathways.**

**(A)** Strategies for simulating the consequence of perturbing transcription factors and signaling pathways. We took advantage of the well-developed CellOracle algorithm to predict the consequence of knocking out transcription factors. For simulating knocking out of signaling pathways, we permutated the expression level of genes in the pathway and then took advantage of TOSICA to re-predict the cell type (this procedure is named as a modified TOSICA in the study). **(B, C)** In silico

### Cell fate bias and $P_{hc}$

Using the probability values of each cell calculated by TOSICA for each cell type, we conducted a subsequent analysis for measuring the cell fate bias. Two cell types are selected for calculation, referred to as cell types A and B. Two equations were described in the main text for capturing the cell fate bias value and $P_{hc}$ value.

### Analyses of differentially expressed genes (DEGs)

We calculated DEGs for each cell type in each of the two groups in four conditions. In the DEG analyses, the FindMarkers function was used from the Seurat package. The Manhattan chart is used to show DEGs between various pairwise comparisons.

### GSEA

We filtered DEGs of certain cell types between two groups. We applied the Seurat FindMarkers function to the log-normalized counts using the MAST method. In addition, we tested all genes expressed in each sample, but only cell types with at least 25 cells per condition were used. We ranked the genes according to their average log fold change. We performed GSEA using the fgsea R package and tested for hallmark gene sets.

### Analysis of transcription factors and their regulons by SCENIC

The SCENIC pipeline was executed using the Python package pySCENIC (46, 85, 86). Firstly, the log-normalized count matrix was used as input, along with a curated list of known TFs, to generate regulons based on their correlation with putative target genes. Secondly, by integrating the generated adjacency matrix with mouse cisTarget databases (10 kbp Up 10 kbp Down and TSS ± 10 kbp), the regulons were refined through pruning targets that lacked enrichment for the corresponding TF motif. Lastly, cells were scored for each regulon with a measure of recovery of target genes from a given regulon.

### Calculation and visualization of pseudotime data

After transferring the Seurat object to the proper AnnData format, we then ran the Python package Palantir with default parameters (60). Only HSCs, GMP, and Pro_NE were used for Palantir analysis for simplicity. We randomly selected an HSPC as the starting point for trajectory analysis. The pseudotime state and gene expression matrix of each cell per cell type or per group were extracted from the simulated datasets and then visualized by ggplot2.

### In silico perturbation of transcription factors by CellOracle

To reduce the computational burden, we scaled the cell number of bone marrow Lin⁻ cells of *Tet2*mut_DSS mice, and 2,500 cells with a total of 15, 991 genes were included in the study. CellOracle (50) was used to predict gene mutation perturbation. The calculated perturbation scores have been shown in the vector form in UMAP. The positive inner product is shown in green (developmental trajectory is strengthened or expedited), whereas the negative inner product is shown in purple (developmental trajectory is reversed).

### In silico perturbation of signaling pathways by a mTOSICA

To predict the consequence of signaling perturbation, a new Seurat expression matrix was constructed by setting the expression of genes belonging to the specific signaling pathway to a null value zero. TOSICA was then applied to annotate the cell type and compare changes in the fraction of cell types, cumulative distribution function, and cell fate bias as described above.

## Data Availability

The raw sequencing data from this study have been deposited in the Genome Sequence Archive in BIG Data Center (https://bigd.big.ac.cn/), Beijing Institute of Genomics (BIG), Chinese Academy of Sciences, under the accession number: PRJCA016651 (the datasets will be available to the public once the article is accepted).

### Code availability

The code and tutorials of the scPlasticity pipeline are available at GitHub: https://github.com/cailab-bio/scPlasticity.

## Acknowledgements

We thank members of Cai Laboratory and colleagues of Tianjin Medical University for their technical and administration support, critics, and helpful suggestions to improve the article. This work was supported in part by grants from the Tianjin Medical University Talent Program and from the National Science Foundation of China to Z Cai (No. 82170173, No. 82371789).

---

perturbations of transcription factors with CellOracle. Perturbation of *Egr1* in *Tet2*mut_Veh, WT_DSS, and *Tet2*mut_DSS is shown in (B). Perturbation of *Egr2*, *Erg*, *Hlf*, and *Meis1* in *Tet2*mut_DSS is shown in (C), respectively. The perturbation score (the inner product) is calculated based on comparing the vectors and visualized by the dots on the grid. The positive inner product is shown in green (developmental trajectory is strengthened and expedited), whereas the negative inner product is shown in red (developmental trajectory is reversed). See the Materials and Methods section or the tutorials by Kamimoto et al for details (https://github.com/morris-lab/CellOracle). See https://github.com/cailab-bio/scPlasticity for our computational scripts. **(D)** In silico perturbations of signaling pathways with a modified TOSICA. We permutated the expression level of genes associated with the indicated signaling pathways to 0 and used TOSICA to re-predict cell types. Stacking bar plots of cell ratios are shown. The ratio values of HSCs, MEP, GMP, Pro_NE, and Pro_Mast are marked. **(E)** Bar plots showing the values of $P_{hc}$ in the indicated hybrid-fate choices. If the $P_{hc}$ value of a perturbation is between that of WT_Veh and *Tet2*mut_Veh, we judge that the perturbation reaches a "partially reversed" effect (marked with #). **(F)** Cumulative density plots in indicated hybrid-fate choices from various scenarios. Similar to (E), if the density value of a perturbation (Prob. = 0.5) is between that of WT_Veh and *Tet2*mut_Veh, we judge that the perturbation reaches a "partially reversed" effect (marked with #).

## Author Contributions

Y Wen: data curation, experimentation, formal analysis, validation, visualization, and writing—original draft.
H He: data curation and experimentation.
Y Ma: data curation.
D Bao: data curation.
LC Cai: data curation and writing—review and editing.
H Wang: resources.
Y Li: resources.
B Zhao: resources.
Z Cai: conceptualization, computing frame and math equation building, formal analysis, funding acquisition, visualization, and writing—review and editing.

## Conflict of Interest Statement

Z Cai is a scientific consultant to Beijing SeekGene BioSciences Co. Ltd. The other authors declare no potential conflict of interest.

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
