## [Reviewer comments · Life Science Alliance]

Life Science Alliance

Computing hematopoiesis plasticity in response to genetic mutations and environmental stimulations

Yuchen Wen, Hang He, Yunxi Ma, Dengyi Bao, Lorie Cai, Huaquan Wang, Yanmei Li, Baobing Zhao, and Zhigang Cai
DOI: <https://doi.org/10.26508/lsa.202402971>

Corresponding author(s): Zhigang Cai, Tianjin Medical University

Review Timeline:

Submission Date:	2024-08-01
Editorial Decision:	2024-10-11
Revision Received:	2024-11-03
Editorial Decision:	2024-11-04
Revision Received:	2024-11-06
Accepted:	2024-11-06

Transaction Report:

October 11, 2024

Re: Life Science Alliance manuscript #LSA-2024-02971-T

Prof. Zhigang Cai
Tianjin Medical University
No22.Qixiangtai Rd.,Heping Dist,Tianjin P.R. China
Tianjin 300070
China

Dear Dr. Cai,

Thank you for submitting your manuscript entitled "Computing hematopoiesis plasticity in response to genetic mutations and environmental stimulations" to Life Science Alliance. The manuscript was assessed by expert reviewers, whose comments are appended to this letter. We invite you to submit a revised manuscript addressing the Reviewer comments.

Thank you for this interesting contribution to Life Science Alliance. We are looking forward to receiving your revised manuscript.

Sincerely,

B. MANUSCRIPT ORGANIZATION AND FORMATTING:

Reviewer #1 (Comments to the Authors (Required)):

The authors theoretically define cell plasticity based on two assumptions (#1 hybrid cells have an equal probability of being in state A or B, and #2 that the proportion of hybrid cells represents plasticity). The argument of the authors is very logical, their assumptions make sense in the light of what we know about hematopoietic stem cells. The study presents a very well argued and well presented application of a theoretical definition of plasticity that the authors defined. The field of plasticity generally lacks a ground truth data to validate the findings, in this sense any work on plasticity will be (for now) mostly descriptive, and I think this study adds an intriguing and conceptually convincing (because of its simplicity) framework to think about it. It generates interesting hypothesis that may help the field understand the impact of TET2. Overall, well done (and no need to oversell it in the discussion)

Comments:

1) the definition of plasticity depends on two arbitrary thresholds (40%-60% proportion of A/B). It would be good to show that this threshold is robust, i.e. how do the results change when 35-65%, or 45-55%? Do the major conclusions hold?

2) I don't think this sentence is necessary: "To our knowledge, however, until the study presented here, there is no pipeline or computational framework to quantify cell plasticity in hematopoiesis and to measure to what extent the plasticity in hematopoiesis responds to gene mutation or environmental stimulation."

There has been other papers describing similar concepts including: <https://pubmed.ncbi.nlm.nih.gov/35260165/> that called it entropy, rather than plasticity.

3) The Snapdragon pipeline should be made available for others to use it (i.e. userfriendly and open access).

Reviewer #2 (Comments to the Authors (Required)):

The authors introduce a new working pipeline for scRNAseq data analysis to interrogate cell states of HSPCs. My main concern with the manuscript is the use of plasticity in this context. Stem cell plasticity would refer to the ability of a stem cell to cross lineage boundaries to acquire the morphological and functional characteristics of a different lineage that is outside of the differentiation lineages they are destined to differentiate into. Thus, in the case of HSPCs to become something different than a hematopoietic cell. In the work in this manuscript they rather investigate the potency of cells; cells that are at the transition of two different hematopoietic lineages and still have the potency to become either of these, depending on the signals they will receive.

- L231: it is not clear what is meant with 'level of plasticity at the transition phase'

- L272: what is meant with 'ambiguity' and how this is defined /determined is not explained well

- L313: a GMP would be most ambiguous in cell fate choice and have the highest degree of plasticity. Could an alternative explanation not be that GMPs are not one well defined cell population, but rather a heterogenous cluster of slightly different cells?

The data analysis with Tet2 mutated cells and DSS treated cells are interesting. However, the results from the analysis do not reveal many new things. IFN and TGFb for example have already been linked to Tet2 mutated malignancies, as has IL1 signaling. Without further validation of these observations, for example by lineage tracing or genetic intervention studies in mice, these data do not add much novelty to what has already been described.

We the authors thank the Editor and Reviewers very much for the constructive comments and giving us the opportunity to revise the above work. We have now revised our manuscript according to the reviewers' comments. The responses to each question and comment are listed point-by-point in the attached pages. We hope that these changes make the revised manuscript acceptable for publication.

By the way, for precisely introducing our pipeline in the revised manuscript and letting the community benefit from our computational frame work, we change the name of the pipeline as scPlasticity (plasticity analysis for single-cell datasets), rather than the fun name SnapDragon. We wish the Editor and Reviwers agree and allow us to do so. Upon the suggestion of Reviewer #1, we also uploaded our computation scripts on the github website (<https://github.com/cailab-bio/scPlasticity>).

Changes in the revised manuscript are highlighted in blue.

Thank you again for your consideration of our manuscript on computing cell plasticity using diversified hematopoietic scRNA-seq inputs.

Response to reviewers

Reviewer #1 (Comments to the Authors (Required)):

The authors theoretically define cell plasticity based on two assumptions (#1 hybrid cells have an equal probability of being in state A or B, and #2 that the proportion of hybrid cells represents plasticity). The argument of the authors is very logical, their assumptions make sense in the light of what we know about hematopoietic stem cells. The study presents a very well argued and well presented application of a theoretical definition of plasticity that the authors defined. The field of plasticity generally lacks a ground truth data to validate the findings, in this sense any work on plasticity will be (for now) mostly descriptive, and I think this study adds an intriguing and conceptually convincing (because of its simplicity) framework to think about it. It generates interesting hypothesis that may help the field understand the impact of TET2. Overall, well done (and no need to oversell it in the discussion)

Response: We the authors appreciate the reviewer's recognition and appreciation of our work. Our work has established an analysis pipeline for effectively and quantitatively quantifying changes in cell plasticity under naïve and stress hematopoiesis. Although our computational frame work is based on many other bio-computational software, we developed a neat and easy-to-follow pipeline and statistical tools to measure cell plasticity with scRNA-seq datasets as inputs. The scripts is now shared on the github website <https://github.com/cailab-bio/scPlasticity>.

By the way, we revised our discussion to compare our pipeline with other platforms computing cell-states and cell-entropy etc.

Comments:

1) the definition of plasticity depends on two arbitrary thresholds (40%-60% proportion of A/B). It would be good to show that this threshold is robust, i.e. how do the results change when 35-65%, or 45-55%? Do the major conclusions hold?

Response: We the authors appreciate the reviewer's comments very much. We define a cell within the range [0.4, 0.6] as a "mathematically reasonable" hybrid cell. The closer the calculated cell fate bias is to 0.5, the more balanced/hybrid the cell's state is between the two cell types.

To test if our conclusions still hold, we have now included the P_{hc} outcome with range [0.3, 0.7], and [0.45, 0.55] in the Table 1 of the revised manuscript. As you can see, our conclusions are still solid.

2) I don't think this sentence is necessary: "To our knowledge, however, until the study presented here, there is no pipeline or computational framework to quantify cell plasticity in hematopoiesis and to measure to what extent the plasticity in hematopoiesis responds to gene mutation or environmental stimulation."

There has been other papers describing similar concepts including: <https://pubmed.ncbi.nlm.nih.gov/35260165/> that called it entropy, rather than plasticity.

Response: We the authors appreciate the reviewer's comments very much. According to our literature tracking, few literatures are **quantitatively** talking about cell plasticity. Among the rare literatures, the pipeline Capybara introduced by Kong et al. touch the concept of hybrid cell but they did not quantify the portion of the hybrid cell along a developmental trajectory(Kong, Fu et al., 2022). Thus, we think we provide a first quantitative way to measure hybrid cells in the developmental process. Anyway, we have now removed the sentences mentioned by the reviewer and also soften our tone in the discussion.

For the study described in <https://pubmed.ncbi.nlm.nih.gov/35260165/>, Charles Dussiau et al.

employed Shannon entropy to calculate the intercellular Shannon entropy for each gene to measure cell-cell variance (Dussiau, Boussaroque et al., 2022). Typically, it is not the concept CELL PLASTICITY we are interested in the manuscript. In our recent clinical work, we also conducted a Palantir-based computational analysis in MDS, a myeloid neoplasm, as described by Charles Dussiau et al (Guo, Jin et al., 2024). It would be interesting to apply the scPlasticity framework to all of the available MDS scRNA-seq datasets. We focused only animal models with Tet2 mutation and DSS treatment here and we will report the outcome of the clinical studies in future.

By the way, in addition to Entrophy, we would argue that a new platform named as “mellon” is closer to our “computing cell plasticity” concept (Otto, Jordan et al., 2024). We have included these two new works in our references and in our revised Discussion. Page 10-11 of the Main text.

3) The Snapdragon pipeline should be made available for others to use it (i.e. userfriendly and open access).

Response: We the authors appreciate the reviewer’s comments very much. We absolutely agree with the reviewer and would like others follow our pipelines to assist their work. We have now uploaded our pipe on the github website: <https://github.com/cailab-bio/scPlasticity>.

By the way, as explained in the cover letter, we renamed the pipeline as scPlasticity to let more people easily identify our motivations and our working flow.

Reviewer #2 (Comments to the Authors (Required)):

The authors introduce a new working pipeline for scRNAseq data analysis to interrogate cell states of HSPCs. My main concern with the manuscript is the use of plasticity in this context. Stem cell plasticity would refer to the ability of a stem cell to cross lineage boundaries to acquire the morphological and functional characteristics of a different lineage that is outside of the differentiation lineages they are destined to differentiate into. Thus, in the case of HSPCs to become something different than a hematopoietic cell. In the work in this manuscript they rather investigate the potency of cells; cells that are at the transition of two different hematopoietic lineages and still have the potency to become either of these, depending on the signals they will receive.

Response: We the authors thank the reviewers very for his/her comments of our work. Indeed, “plasticity” is a very old concept of developmental biology, especially for Stem cells. Before the application of scRNA-seq dataset, people solely rely on experimentation to test “cell plasticity” in a developmental process. However, the availability of scRNA-seq make it possible to capture the “cell plasticity” and “developmental continuum” even just based on a “snapshot” dataset if we image that mature cells in the datasets could come from the progenitor cells in the same dataset. This is the overall hypothesis of developmental biologist who are using scRNA-seq for their biological or computational work. Yes, hematopoiesis dataset from the bone marrow cells, especially the Lin-negative cells described here, perfectly serves as the “snapshot” dataset analyzed in the study.

As addressed in the introduction of the original version, we define “cell plasticity” solely on cell-fate choice and with inputs from scRNA-seq dataset. We did not touch the inputs from cell morphology, activity, or other parameters.

We provide Assumptions #1 and #2, Equation #1 and #2, several statistical measurements (confusion-matrix-like heatmap, cell fate bias heatmap etc) and shared our computational scripts on the github websit. Thus, we believe our definition of “cell plasticity” and our pipeline is clear,

simple and biologically interpretable.

- L231: it is not clear what is meant with 'level of plasticity at the transition phase'

Response: We the authors appreciate the reviewer's comments very much. As we know, development is a continuous process of various cell fates, however, the Seurat-based cell types we defined in single-cell transcriptome analysis are discrete and deterministic. We have to rely on other parameters to capture the cell plasticity (CP). We manually define the portion of hybrid cell (P_{hc}) as a measure of CP. 'Level of plasticity at the transition stage' means that we care about the range of plasticity is in the group of cells that are in the transition stage, Therefore, the proportion of hybrid cells (cell fate bias between 0.4 and 0.6) was calculated to represent the plasticity of this group of cells in the transition state.

- L272: what is meant with 'ambiguity' and how this is defined /determined is not explained well

Response: We the authors appreciate the reviewer's comments very much. The concept "ambiguity" refers to "all of the possibility" values of a cell as a cell-fate HSC, GMP, MEP etc. The cell type annotated by Seurat, delineate clear boundaries, representing a "fixed" state. However, when query with TOSICA, "cell-type/cell-fate/cell-annotation" ambiguity will show up and could be statistically measured by various plots or calculations as shown in **Figure 2**.

- L313: a GMP would be most ambiguous in cell fate choice and have the highest degree of plasticity. Could an alternative explanation not be that GMPs are not one well defined cell population, but rather a heterogenous cluster of slightly different cells?

Response: We the authors appreciate the reviewer's comments very much. His/her understanding of GMP is absolutely right. GMP cells annotated in the UMAP plots are based on the markers of experimentally defined GMP (flow cytometry defined and other experimental work). We named it as GMP just for following the same concept from the experimental hematologists. Both Mathematically and Biologically, GMP compartments have many subgroups for sure. We did not subcluster the GMP pool as the concept GMP is easily connected to the traditional understanding of in-between cell types (Between HSC and Pro_NE).

The data analysis with Tet2 mutated cells and DSS treated cells are interesting. However, the results from the analysis do not reveal many new things. IFN and TGFb for example have already been linked to Tet2 mutated malignancies, as has IL1 signaling. Without further validation of these observations, for example by lineage tracing or genetic intervention studies in mice, these data do not add much novelty to what has already been described.

Response: We the authors appreciate the reviewer's comments very much. The novelty of the study is certainly not about the biological findings from our scRNA-seq dataset. The The novelty of the study is purely relying on our computational pipeline scPlasticity (the old name SnapDragon was removed as we explained in the cover letter). Any novel clues from our analysis certainly require further functional verification and it will be explored in our future study.

More importantly, as we outlined in the pipeline and detailed in the **Figure 7D-F**, we could be able to *in silico* simulate the outcome of perturbing signaling pathways with the modified TOSICA framework (mTOSICA). Thus, both transcriptional factors and signaling pathways would be simulated in our framework scPlasticity. We wish the reviewer could the appreciate our endeavor of our study as a complete and interesting story.

References:

Dussiau, C., Boussaroque, A., Gaillard, M., Bravetti, C., Zaroili, L., Knosp, C., Friedrich, C., Asquier, P., Willems, L., Quint, L., et al. (2022). Hematopoietic differentiation is characterized by a transient peak of entropy at a single-cell level. *BMC Biol* 20, 60. doi:10.1186/s12915-022-01264-9.

Guo, X., Jin, W., Wen, Y., Wang, Z., Ren, X., Liu, Z., Fu, R., Cai, Z., and Li, L. (2024). Computing cell state discriminates the aberrant hematopoiesis and activated microenvironment in Myelodysplastic syndrome (MDS) through a single cell genomic study. *J Transl Med* 22, 673. doi:10.1186/s12967-024-05496-x.

Kong, W., Fu, Y.C., Holloway, E.M., Garipler, G., Yang, X., Mazzoni, E.O., and Morris, S.A. (2022). Cappybara: A computational tool to measure cell identity and fate transitions. *Cell Stem Cell* 29, 635-649. e611.

Otto, D.J., Jordan, C., Dury, B., Dien, C., and Setty, M. (2024). Quantifying cell-state densities in single-cell phenotypic landscapes using Mellon. *Nat Methods* 21, 1185-1195. doi:10.1038/s41592-024-02302-w.

November 4, 2024

RE: Life Science Alliance Manuscript #LSA-2024-02971-TR

Prof. Zhigang Cai
Tianjin Medical University
No22.Qixiangtai Rd., Heping Dist, Tianjin P.R. China
Tianjin 300070
China

Dear Dr. Cai,

Thank you for submitting your revised manuscript entitled "Computing hematopoiesis plasticity in response to genetic mutations and environmental stimulations". We would be happy to publish your paper in Life Science Alliance pending final revisions necessary to meet our formatting guidelines.

- please be sure that the authorship listing and order is correct
- please add your ORCID ID for the corresponding author-you should have received instructions on how to do so
- please add the Twitter handle of your host institute/organization as well as your own or/and one of the authors in our system
- please upload your table files as editable doc or excel file
- please make the raw sequencing data publicly accessible at this point

Figure Check:

- your Figure 4 figure legend has the panels A-I, but the panels G-I are not in the figure and are not called out in the text; please correct

A. FINAL FILES:

B. MANUSCRIPT ORGANIZATION AND FORMATTING:

Sincerely,

November 6, 2024

RE: Life Science Alliance Manuscript #LSA-2024-02971-TRR

Prof. Zhigang Cai
Tianjin Medical University
No22.Qixiangtai Rd.,Heping Dist,Tianjin P.R. China
Tianjin 300070
China

Dear Dr. Cai,

Thank you for submitting your Methods entitled "Computing hematopoiesis plasticity in response to genetic mutations and environmental stimulations". It is a pleasure to let you know that your manuscript is now accepted for publication in Life Science Alliance. Congratulations on this interesting work.

DISTRIBUTION OF MATERIALS:

Again, congratulations on a very nice paper. I hope you found the review process to be constructive and are pleased with how the manuscript was handled editorially. We look forward to future exciting submissions from your lab.

Sincerely,
